

**Exclusively heteronuclear NMR experiments for the investigation of intrinsically disordered**
**proteins: focusing on proline residues**
Isabella C. Felli[1], Wolfgang Bermel[2] and Roberta Pierattelli[1]
[1]CERM Department of Chemistry ''Ugo Schiff'', University of Florence, Via Luigi Sacconi 6, 50019 Sesto
Fiorentino, Florence, Italy
[2]Bruker BioSpin GmbH, Silberstreifen, 76287 Rheinstetten, Germany
*Corresponce to:*
Isabella C. Felli (felli@cerm.unifi.it) and Roberta Pierattelli (roberta.pierattelli@unifi.it)
**Keywords**
$^{13}$C detection, IDP, NMR
**Abstract**
NMR represents a key spectroscopic technique to contribute to the emerging field of highly flexible,
intrinsically disordered proteins (IDPs) or protein regions (IDRs) that lack a stable three-dimensional
structure. A set of exclusively heteronuclear NMR experiments tailored for proline residues, highly
abundant in IDPs/IDRs, are presented here. They provide a valuable complement to the widely used
approach based on amide proton detection, filling the gap introduced by the lack of amide protons in
prolines within polypeptide chains. The novel experiments have very interesting properties for the
investigations of IDPs/IDRs of increasing complexity.




## Introduction

Invisible in X-ray studies of protein crystals, intrinsically disordered regions (IDRs) of complex proteins have been for a long time considered just passive linkers connecting functional globular domains and thus often ignored in structural biology studies. However, in many cases they comprise a significant fraction of the primary sequence of a protein and for this reason they are expected to have a role in protein function (Van Der Lee et al., 2014). The characterization of highly flexible regions of large proteins as well as entire proteins characterized by the lack of a 3D structure, now generally referred to as intrinsically disordered proteins (IDPs), lies well behind that of their folded counterparts and is nowadays pursued by an increasingly large number of studies to fill this knowledge gap. NMR plays a strategic role in this context since it constitutes the major, if not the unique, spectroscopic technique to achieve atomic resolution information on their structural and dynamic properties. However, intrinsic disorder and high flexibility have very relevant effects for NMR investigations such as reduction of chemical shift dispersion as well as efficient exchange processes with the solvent due to the open conformations that, when approaching physiological pH and temperature, broaden amide proton resonances beyond detection. While several elegant experiments were proposed to exploit exchange processes with the solvent (Kurzbach et al., 2017; Olsen et al., 2020; Szekely et al., 2018; Thakur et al., 2013), in general initial NMR investigations of IDPs/IDRs are carried out in conditions in which these critical points are mitigated. Exchange broadening strongly depends on pH and temperature; conditions can be optimized to recover most of the amide proton resonances enabling the acquisition of amide proton detected triple resonance experiments needed for sequence-specific assignment of the resonances. However, in particular for proteins that are largely exposed to the solvent it may be interesting to study them near physiological pH and temperature conditions (Gil et al., 2013). In this context, $^{13}$C direct detection NMR developed into a valuable alternative.

Although the intrinsic sensitivity of $^{13}$C is lower with respect to that of $^1$H, $^{13}$C nuclear spins are characterized by a large chemical shift dispersion (Dyson, H. Jane; Wright, 2001) and, when coupled to $^{15}$N nuclei, provide a well-defined fingerprint of a polypeptide (Bermel et al., 2006a; Hsu et al., 2009; Lopez et al., 2016; Schiavina et al., 2019). These features were exploited to design a suite of 3D experiments based on carbonyl-carbon direct detection for sequential assignment and to measure NMR observables (Felli and Pierattelli, 2014). These experiments exploit only heteronuclear chemical shifts in the indirect dimensions to maximize chemical shift dispersion (exclusively heteronuclear experiments) and can be used to study IDPs/IDRs also in conditions in which amide proton resonances are too broad to be detected. In addition, they reveal information about proline residues that lack the amide proton when part of



polypeptide chains and cannot be detected in 2D HN correlation experiments even if pH and temperature
conditions are optimized to reduce exchange broadening.
Proline residues are abundant in IDPs/IDRs and often occur in proline-rich sequences with repetitive units
(Theillet et al., 2014). Initial bioinformatics studies on the relative abundance of each amino acid in regions
of the protein that could not be observed in X-ray diffraction studies led to the classification of prolines
as "disorder promoting" amino acids (Dunker et al., 2008). Nevertheless proline, the only imino acid,
features a closed ring in its side chain which confers local rigidity compared to all other amino acids
(Williamson, 1994), as also exploited in FRET studies in which proline residues are used as rigid spacers to
measure distances (Schuler et al., 2005). These observations clearly show the importance of experimental
atomic resolution information on the structural and dynamic properties of proline residues to understand
their role in modulating protein function. While abundant information about proline residues in globular
protein folds is available either through NMR or X-ray studies (MacArthur and Thornton, 1991), including
several examples of cis-trans isomerization of peptide bonds involving proline nitrogen as molecular
switches (Lu et al., 2007), their characterization in highly flexible, disordered polypeptides is available only
in a handful of cases (Chaves-Arquero et al., 2018; Gibbs et al., 2017; Haba et al., 2013; Hošek et al., 2016;
Knoblich et al., 2009; Pérez et al., 2009; Piai et al., 2016) and actually early studies on IDPs/IDRs routinely
reported assignment statistics only considering all other amino acids ("excluding prolines").
Here we would like to propose an experimental variant of the most widely used [13]C detected 3D
experiments for sequence-specific assignment of IDPs/IDRs to selectively pick up correlations involving
proline nitrogen nuclei and provide key complementary information to that obtained through amide
proton detected experiments. They can be collected in a shorter time with respect to standard 3D
experiments and provide a valuable addition to the current experimental protocols for the study of IDPs.

**Materials and methods**
Isotopically labelled α-synuclein ([13]C and [15]N) was expressed and purified as previously described (Huang
et al., 2005). The NMR sample has 0.6 mM protein concentration in 20 mM phosphate buffer at pH 6.5
and 100 mM NaCl in $H_2O$ with 5% $D_2O$ for the lock signal.
Isotopically labelled CBP-ID4 ([13]C and [15]N) was expressed and purified as previously described (Piai et al.,
2016). The NMR sample has 0.9 mM protein concentration in water buffer containing 20 mM TRIS, 50 mM
KCl, at pH 6.9, with 5% $D_2O$ added for the lock signal.



NMR experiments were acquired at 288 K (for α-synuclein) and at 283K (for CBP-ID4) with a 16.4 T Bruker
AVANCE NEO spectrometer operating at 700.06 MHz $^1$H, 176.05 MHz $^{13}$C, and 70.97 MHz $^{15}$N frequencies,
equipped with a 5 mm cryogenically cooled probehead optimized for $^{13}$C direct detection (TXO). RF pulses
and carrier frequencies typically employed for the investigation of intrinsically disordered proteins were
used, except for the modifications introduced to zoom into the proline $^{15}$N region. Carrier frequencies
were set to 4.7 ppm ($^1$H), 176.4 ($^{13}$C'), 53.9 ($^{13}$C$^\alpha$), 44.9 ($^{13}$C$^{ali}$); the $^{15}$N carrier was set to 137 ppm, in the
center of $^{15}$N resonances of proline residues. Hard pulses were used for $^1$H. Band selective $^{13}$C pulses used
were Q5 and Q3 (Emsley and Bodenhausen, 1990) of 300 μs and 231 μs for 90° and 180° rotations
respectively; a 900 μs Q3 pulse centered at 53.9 ppm was used for selective inversion of C$^\alpha$. The $^{15}$N pulse
to invert the $^{15}$N proline resonances was a 8000 μs Reburp pulse (Geen and Freeman, 1991); all other $^{15}$N
pulses were hard pulses. Decoupling was achieved with waltz65 (100 μs, 2.5 kHz) (Zhou et al., 2007) for
$^1$H and with garp4 (250 μs, 1.0 kHz) (Shaka, A. J.; Barker, P. B.; Freeman, 1985) for $^{15}$N. The MOCCA mixing
time (Felli et al., 2009; Furrer et al., 2004) in the (HCA)COCON$^{Pro}$ experiment was 350 ms, constituted by
repeated ($\Delta$-180°-$\Delta$)$_{2n}$ units in which $\Delta$=150 μs and the 180° pulse was 91.6 μs).
The experimental parameters used for the acquisition of the various experiments on α-synuclein and CBP-
ID4 are reported in Table 1. Spectra were calibrated using DSS as a reference for $^1$H and $^{13}$C; $^{15}$N was
calibrated indirectly (Markley et al., 1998).



**Table 1.** Experimental parameters used.

| Experiments α-synuclein | Dimension of acquired data | | | Spectral width (ppm) | | | NS[a] | $d_1(s)$[b] |
|---|---|---|---|---|---|---|---|---|
| | **t1** | **t2** | **t3** | **F1** | **F2** | **F3** | | |
| **$^1$H detected** | | | | | | | | |
| $^1$H-$^{15}$N HSQC | 800 ($^{15}$N) | 2048 ($^1$H) | | 28.1 | 15.0 | | 2 | 1.0 |
| **$^{13}$C detected** | | | | | | | | |
| CON | 512 ($^{15}$N) | 1024 ($^{13}$C) | | 32.0 | 31.0 | | 2 | 1.6 |
| CON$^{Pro}$ | 128 ($^{15}$N) | 1024 ($^{13}$C) | | 5.0 | 31.0 | | 2 | 1.6 |
| (H)CBCACON$^{Pro}$ | 128 ($^{13}$C) | 32 ($^{15}$N) | 1024 ($^{13}$C) | 60.0 | 5.0 | 30.0 | 4 | 1.0 |
| (H)CCCON$^{Pro}$ | 128 ($^{13}$C) | 32 ($^{15}$N) | 1024 ($^{13}$C) | 70.0 | 5.0 | 30.0 | 4 | 1.0 |
| (H)CBCANCO$^{Pro}$ | 128 ($^{13}$C) | 16 ($^{15}$N) | 1024 ($^{13}$C) | 60.0 | 5.0 | 30.0 | 8 | 1.0 |
| **$^1$H and $^{13}$C detected (using multiple receivers)** | | | | | | | | |
| CON/HN | 600 ($^{15}$N) | 1024 ($^{13}$C) | | 35.0 | 31.0 | | 2 | 1.6 |
| | 600 ($^{15}$N) | 2048 ($^1$H) | | 35.0 | 15.0 | | 4 | |
| [a] number of acquired scans | | | | | | | | |
| [b] inter-scan delay | | | | | | | | |
| Experiments CBP-ID4 | Dimension of acquired data | | | Spectral width (ppm) | | | NS[a] | $d_1(s)$[b] |
| | **t1** | **t2** | **t3** | **F1** | **F2** | **F3** | | |
| **$^1$H detected** | | | | | | | | |
| $^1$H-$^{15}$N HSQC | 800 ($^{15}$N) | 2048 ($^1$H) | | 30.0 | 15.0 | | 2 | 1.0 |
| **$^{13}$C detected** | | | | | | | | |
| CON | 1024 ($^{15}$N) | 1024 ($^{13}$C) | | 38.0 | 30.0 | | 2 | 2.0 |
| CON$^{Pro}$ | 170 ($^{15}$N) | 1024 ($^{13}$C) | | 6.5 | 30.0 | | 2 | 2.0 |
| (H)CBCACON$^{Pro}$ | 128 ($^{13}$C) | 64 ($^{15}$N) | 1024 ($^{13}$C) | 64.5 | 6.5 | 30.0 | 4 | 1.0 |
| (H)CCCON$^{Pro}$ | 128 ($^{13}$C) | 64 ($^{15}$N) | 1024 ($^{13}$C) | 75.7 | 6.5 | 30.0 | 4 | 1.0 |
| (H)CBCANCO$^{Pro}$ | 128 ($^{13}$C) | 22 ($^{15}$N) | 1024 ($^{13}$C) | 64.5 | 6.5 | 30.0 | 16 | 1.0 |
| (HCA)COCON$^{Pro}$ | 96 ($^{13}$C) | 64 ($^{15}$N) | 1024 ($^{13}$C) | 10.8 | 6.5 | 30.0 | 8 | 1.5 |
| [a] number of acquired scans | | | | | | | | |
| [b] inter-scan delay | | | | | | | | |




**Results and discussion**
*Advantages of focusing on proline residues*
In highly flexible and disordered proteins contributions to signals' chemical shifts deriving from the local
environment are averaged out leaving mainly those contributions due to the covalent structure of the
polypeptide. Chemical shift ranges predicted for $^{15}$N resonances of imino acids such as proline are quite
different from those predicted for amino acids, as expected from the different chemical structure. The 2D
CON spectra of several disordered proteins of different size and sequence complexity, reported in Figure
1, clearly show that proline residues are quite abundant in IDPs/IDRs and that indeed $^{15}$N resonances of
proline residues fall in a well isolated spectral region.

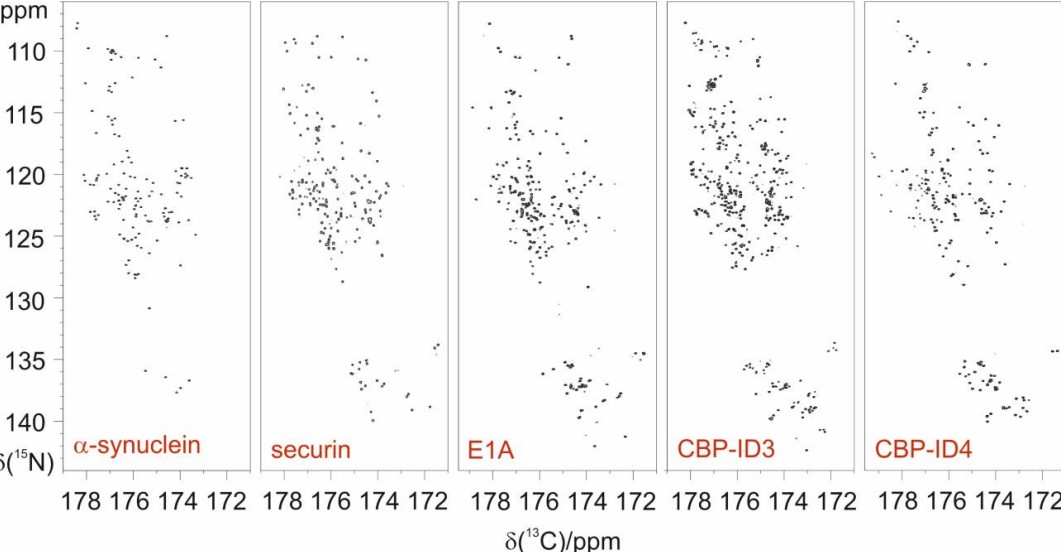


**Figure 1**. Proline residues are abundant in IDPs/IDRs and their $^{15}$N resonances can be easily detected. They
fall in a specific, isolated region of the 2D CON spectrum as illustrated by the examples reported in the
figure. From left to right: α-synuclein (140 aa, 4% Pro)(Bermel et al., 2006b); human securin (200 aa, 11%
Pro)(Bermel et al., 2009); E1A (243 aa, 16% Pro)(Hošek et al., 2016); CBP-ID3 (407 aa 18% Pro)(Contreras-
Martos et al., 2017); CBP-ID4 (207 aa, 22% Pro)(Piai et al., 2016).





Thus, $^{15}$N resonances of proline residues in IDPs/IDRs can be selectively irradiated enabling us to focus on
this spectral region. This can be achieved through the use of band-selective $^{15}$N pulses as shown for the
simple case of the CON experiment (Murrali et al., 2018): the selective CON spectrum in the proline region
(CON$^{Pro}$, Figure 2) provides the complementary information that is missing in 2D HN correlation
experiments, even when pH and temperature are optimized to enhance the detectability of amide protons
(Figure 2).

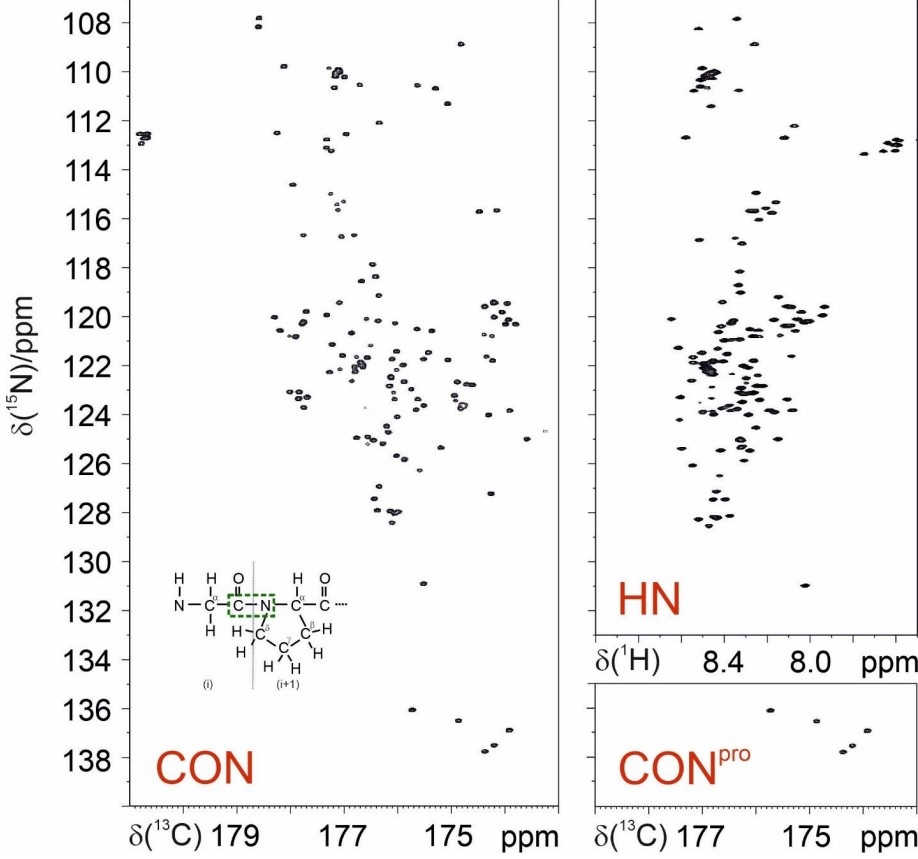



**Figure 2**. Comparison of the 2D CON (left) and 2D HN (right, top) spectra recorded on α-synuclein to
illustrate the larger signals' dispersion in the former. The CON$^{pro}$ spectrum (right, bottom), reported below
the HN panel, clearly illustrates how this experiment provides the missing information with respect to that
available in the HN-detected spectrum. In the inset of the 2D CON spectrum a scheme of a Gly$_{i-1}$-Pro$_i$
dipeptide highlights the nuclei that give rise to the C'$_{i-1}$N$_i$ correlations detected in CON spectra (circled in
green).

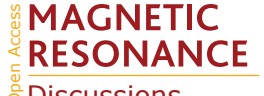

The same strategy exploiting band-selective $^{15}$N pulses can be used to design experimental variants of
triple resonance $^{13}$C detected experiments to focus on the $^{15}$N proline region and enable us to selectively
detect the desired correlations. When implementing this idea into these experiments, such as the 3D
(H)CBCACON (Bermel et al., 2009), $^{15}$N pulses could all be substituted with band-selective ones. However,
instead of substituting all $^{15}$N pulses, it is sufficient to introduce a 180° band-selective $^{15}$N pulse in one of
the $C'_{i-1}$-$N_i$ coherence transfer steps to introduce the desired selectivity in the proline region.
As an example the pulse sequence of the 3D (H)CBCACON$^{Pro}$ experiment is shown in Figure 3. The inclusion
of the $^{15}$N band-selective pulse in the C'-N coherence transfer step is used to generate the $C'_{i-1}$-$N_i$ antiphase
coherence ($2C'_yN_z$) involving the $^{15}$N nuclear spin of proline residues (i); for all other amino acid types the
evolution of the $C'_{i-1}$-$N_i$ scalar coupling ($^1J_{C'i-1Ni}$) is refocused by the 180° band-selective pulse on the
carbonyl carbon nuclei only. To achieve the desired selectivity on the $^{15}$N proline resonances with respect
to those of all other amino acids an 8 ms Reburp pulse (Geen and Freeman, 1991) was used here; this
pulse may appear quite long, but it can be accommodated well in the C'-N coherence transfer block that
requires about 32 ms ($1/2^1J_{C'i-1Ni}$). Considering a 8-10 ppm spectral width necessary to cover the $^{15}$N
proline-region in the indirect dimension (Figure 1), the implementation of this $^{15}$N band-selective pulse
allows us to reduce the spectral width by a factor of about 4 with respect to that needed to cover the
whole spectral region in which backbone $^{15}$N nuclear spins resonate, that is about 36-40 ppm. This means
that the same resolution can be achieved in a fraction of the time since ¼ (or less) of the FIDs should be
collected, provided sensitivity is not a limiting factor. Thus, it becomes feasible to acquire spectra with
very high resolution, extending the acquisition time in all the indirect dimensions to contrast the reduced
chemical shift dispersion typical of IDPs. Non-uniform sampling strategies (Hoch et al., 2014; Kazimierczuk
et al., 2010, 2011; Robson et al., 2019) can of course be implemented to reduce acquisition times; also in
this case reducing the spectral complexity (the number of cross-peaks is reduced when focusing on proline
$^{15}$N resonances only) is expected to contribute to reducing experimental times. Out of the full 3D spectrum
only a small portion, the one containing the information that is completely missing in amide proton
detected experiments, can thus be acquired with the necessary resolution to provide site-specific atomic
information.
Since C' detected experiments all exploit the $C'_{i-1}$-$N_i$ correlation, which is an inter-residue correlation
linking the nitrogen of an amino acid (i) to the carbonyl carbon of the previous one (i-1) across the peptide
bond (Figure 2 inset), focusing on proline residues can facilitate the identification of specific $X_{i-1}$-$Pro_i$ pairs
through inspection of $C^\alpha$ and $C^\beta$ chemical shifts of the amino acid preceding proline residues. Such



information can be achieved through  the 3D (H)CBCACON$^{Pro}$ experiment and can be very useful to identify
specific pairs such as Gly/Pro, Ala/Pro, Ser/Pro and Thr/Pro. Acquisition of the 3D (H)CCCON$^{Pro}$
experiment, in parallel to the 3D (H)CBCACON$^{Pro}$, provides information on aliphatic $^{13}$C nuclear chemical
shifts of the whole side chain. This contributes to narrowing down the possibilities in all cases in which it
is not possible to identify the type of amino acid preceding the proline only considering their $^{13}$C$^{\alpha}$ and $^{13}$C$^{\beta}$
chemical shifts. This is the case for example of Arg/Lys/Gln or Phe/Leu.
Similarly, the insertion of the $^{15}$N band-selective pulse in the proline region in the 3D (H)CBCANCO (Bermel
et al., 2006a) enables us to detect the $^{13}$C resonances of the whole proline ring, providing the
complementary information for sequence-specific assignment. The closed proline ring introduces an
additional heteronuclear scalar coupling ($^{1}J_{Ni-C\delta i}$) that also provides the correlations with $^{13}$C$^{\delta}$ and $^{13}$C$^{\gamma}$, in
parallel to $^{13}$C$^{\alpha}$ and $^{13}$C$^{\beta}$ chemical shifts. Indeed the band-selective pulses used for the $^{13}$C aliphatic region
cover also $^{13}$C$^{\gamma}$ and $^{13}$C$^{\delta}$ resonances (not only $^{13}$C$^{\alpha}$ and $^{13}$C$^{\beta}$ ones). In addition analysis of the observed
chemical shifts for proline side chains can be correlated to the local conformation, in particular to the
cis/trans isomers of the peptide bond involving proline nitrogen nuclei (Schubert et al., 2002; Shen and
Bax, 2010). Finally, additional information for sequence-specific assignment can be achieved by exploiting
the same approach for the COCON experiment in its $^{13}$C-start (Bermel et al., 2006b; Felli et al., 2009)  as
well and in its $^{1}$H-start variants (Mateos et al., 2020).

*Assignment strategy*
To illustrate the approach, experiments were acquired on the well-known IDP α-synuclein. Even if this
protein only contains a little number of proline residues (5/140), they are all clustered in a small portion
of it (108-138) and thus constitute about 15% of the amino acids in this region. Furthermore, this terminus
has a very peculiar amino acidic composition (36% Asp/Glu, 9% Tyr) and it was shown to be the part of
the protein that is involved in sensing calcium concentration jumps associated with the transmission of
nerve signals (Binolfi et al., 2006; Lautenschläger et al., 2018; Nielsen et al., 2001). Proline residues,
embedded in two motifs (DPD and EPE), were shown to facilitate the interaction of carboxylate side chains
of Asp and Glu with calcium, even in a flexible, disordered state (Pontoriero et al., 2020).
Focusing on the proline $^{15}$N region in case of α-synuclein greatly simplifies the spectral complexity enabling
us to illustrate the sequence-specific assignment of the resonances just by visual inspection of the first
planes of the 3D spectra described here (Figure 3).



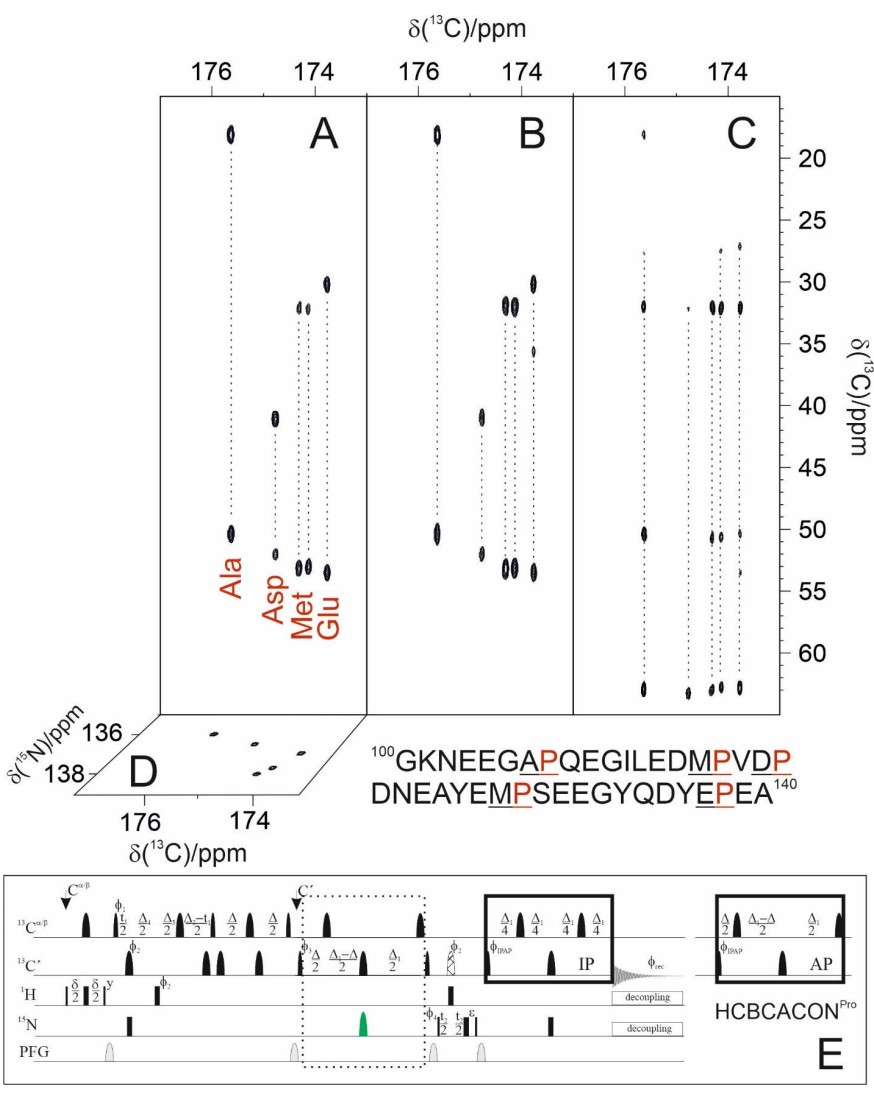

**Figure 3**. The implementation of the proposed strategy on α-synuclein renders NMR spectra so informative that proline resonances can be assigned just by visual inspection of the figure. To this end the first $^{13}C$-$^{13}C$ planes of the 3D (H)CBCACON$^{Pro}$ (A), 3D (H)CCCON$^{Pro}$ (B), 3D (H)CBCANCO$^{Pro}$ (C) are shown; the $^{13}C$-$^{15}N$ plane of the 3D (H)CBCACON$^{Pro}$ is also shown (D). The portion of the primary sequence of α-synuclein hosting its five proline residues is also reported (bottom right). The pulse sequence to acquire the 3D (H)CBCACON$^{Pro}$ experiment (E) is reported as an example of the implementation of the proposed approach (dotted box). The delays are: $\varepsilon = t_2(0)$, $\delta = 3.6$ ms, $\Delta = 9$ ms, $\Delta_1 = 25$ ms, $\Delta_2 = 8$ ms, $\Delta_3 = \Delta_2 - \Delta_4 = 5.8$ ms, $\Delta_4 = 2.2$ ms. The phase cycle is: $\phi_1 = x, -x$; $\phi_2 = 8(x), 8(-x)$; $\phi_3 = 4(x), 4(-x)$; $\phi_4 = 2(x), 2(-x)$; $\phi_{IPAP}(IP) = x$; $\phi_{IPAP}(AP) = -y$; $\phi_{rec} = x, -x, -x, x, -x, x, x, -x$. Quadrature detection was obtained by incrementing phase $\phi_3$ ($t_1$) and $\phi_4$ ($t_2$) in a States-TPPI manner.



Indeed, the C'-N projections of the 3D spectra ($^{13}$C-$^{15}$N planes) show that the cross-peaks are well resolved
in both dimensions; selection of the $^{15}$N proline region enables us to differentiate the signals through the
carbonyl carbon chemical shifts of the preceding amino acid. Therefore inspection of the first $^{13}$C-$^{13}$C plane
of the 3D (H)CBCACON$^{Pro}$ experiment (Figure 3A) shows the distinctive $^{13}$C$^{\alpha}$ and $^{13}$C$^{\beta}$ chemical shift patterns
of the residues preceding proline that, by comparison with the primary sequence of the protein, already
suggest us the identity of three residue pairs: Ala-Pro, Asp-Pro, Glu-Pro. These can thus be assigned to Ala
107-Pro 108, Asp 119-Pro 120, Glu 138-Pro 139. Comparison with the first $^{13}$C-$^{13}$C plane of the 3D
(H)CCCON$^{Pro}$ (Figure 3B) confirms that an extra cross peak can be detected for the Glu-Pro pair, as
expected for amino acids which have a side chain with more than two aliphatic carbon atoms. The
remaining signals derive from the two Met-Pro pairs, in agreement with the observed chemical shifts.
They can be assigned in a sequence specific manner by comparison of these spectra with the
complementary ones based on amide proton detection. The final panel shows the first $^{13}$C-$^{13}$C plane of
the 3D (H)CBCANCO$^{Pro}$ (Figure 3C). This experiment reveals the correlations of the $^{15}$N with $^{13}$C$^{\alpha}$ and $^{13}$C$^{\beta}$
within each amino acid. In the case of proline, the closed ring introduces additional scalar couplings that
are responsible for two additional cross-peaks, the ones of the $^{15}$N with $^{13}$C$^{\gamma}$ and $^{13}$C$^{\delta}$, as clearly observed
in Figure 3. Chemical shifts show only minor differences between the resonances but still significant to
discriminate the different residues provided spectra are acquired with high resolution.

*A challenging case*
A compelling example of complexity is provided by the ID4 flexible linker of CREB-binding protein (CBP), a
large transcription co-regulator (Dyson and Wright, 2016). CBP-ID4 connects two well-characterized
globular domains (TAZ2, 92 amino acids and NCBD, 59 amino acids)(De Guzman et al., 2000; Kjaergaard
et al., 2010) and is constituted by 207 amino acids out of which 45 are proline residues, including several
repeated PP motives (Piai et al., 2016) (Figure 4A). The 2D CON$^{Pro}$ spectrum of CBP-ID4, reported in Figure
4B (left panel), shows the C'$_{i-1}$-N$_i$ correlations of proline residues of ID4. Interestingly, despite the small
spectral region, a high number of resolved resonances is observed. The initial count of cross-peaks in this
spectrum reveals 42 out of the 45 expected correlations, highlighting the potential of this experimental
strategy for the investigation of IDRs/IDPs of increasing complexity. The excellent chemical shift dispersion
of the inter-residue C'$_{i-1}$-N$_i$ correlations is certainly one of the most important aspects to reduce cross peak
overlap. Resolution is further enhanced in this region by the narrow linewidths of proline $^{15}$N resonances
due to the lack of the dipolar contribution of an amide proton to the transverse relaxation.



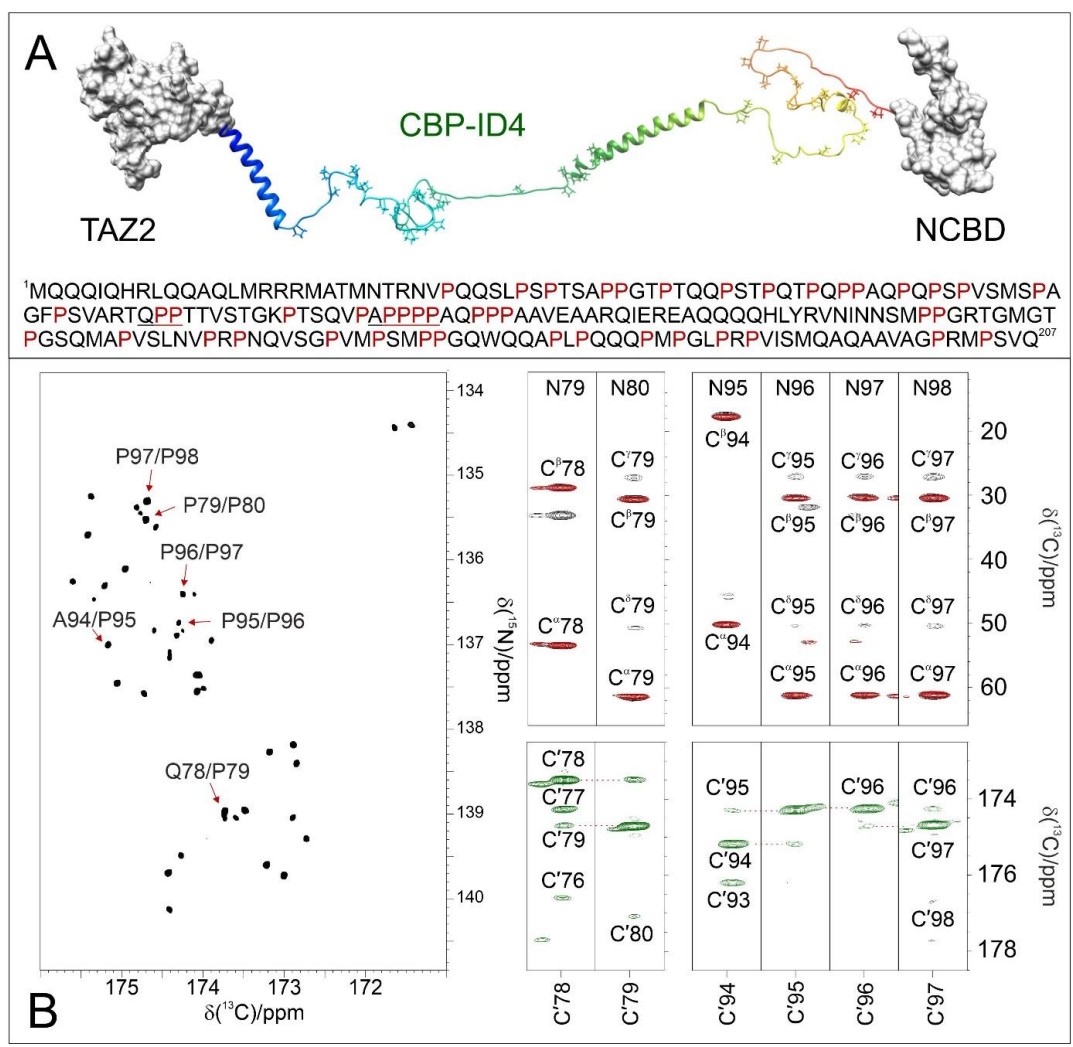

**Figure 4**. The case of CBP-ID4 (residues 1851–2057 of human CBP). (A) One of the possible conformations of the CBP-ID4 fragment (blue-to-red ribbons) and the two flanking domains TAZ2 and NCBD clearly demonstrate that in this region of CBP the intrinsically disordered part is highly prevalent with respect to ordered ones. The amino acid sequence of CBP-ID4 is also reported. (B) The 2D CON$^{Pro}$ spectrum shown on the left, which at first sight could seem like a 2D spectrum of a small globular protein, reports the proline fingerprint of this complex IDR. Several strips extracted from the 3D (H)CBCACON$^{Pro}$ (black contours), 3D (H)CCCON$^{Pro}$ (red contours), 3D (HCA)COCON$^{Pro}$ (green contours) are shown to illustrate the information available for sequence specific resonance assignment of this proline-rich fragment of CBP-ID4.





These two features contribute to establishing this spectral region as a key one for the assignment of a
large IDR. Indeed, when passing from 2D to 3D experiments, long acquisition times in the $^{15}N$ dimension
are possible and enable us to provide the extra contribution to resolution enhancement needed to focus
on complex IDRs and to collect additional information on the proline residues and their neighbouring
amino acids. As an example of the quality of the spectra, Figure 4B reports several strips extracted from
the pro-selective 3D experiments that were essential for the investigation of a particularly proline-rich
region of ID4, the one in between two partially populated α-helices (Piai et al., 2016). This is composed by
27 prolines (out of 76 amino acids) which constitute 35% of the amino acids in this region including several
proline-rich motifs (PXP, PXXP, PP as well as PPP and PPPP). The strips extracted from the 3D
(H)CBCACON$^{Pro}$ and 3D (H)CCCON$^{Pro}$ (Figure 4B, right panels, black and red contours respectively) are very
useful to identify the $X_{i-1}$-Pro$_i$ pairs that match in this case with a Gln-Pro and an Ala-Pro pair as well as
several Pro-Pro ones. The 3D (H)CBCANCO$^{Pro}$ completes the picture providing information about $^{13}C$
resonances of each proline ring (C$^{α}$, C$^{β}$, C$^{γ}$, C$^{δ}$, not shown for sake of clarity of the figure). However in
regions with a high abundance of proline residues additional information is needed for their sequence-
specific assignment. To this end the 3D (HCA)COCON$^{Pro}$ (Figure 4B, right panels, green contours) is very
useful, as demonstrated for these two proline-rich fragments. This experiment, which includes an
isotropic mixing element in the carbonyl region (MOCCA in this case (Felli et al., 2009; Furrer et al., 2004)),
enables us to detect correlations of a carbonyl with the neighbouring ones through the small $^3J_{C'C'}$ scalar
couplings. In case of proline residues the most intense cross-peak is generally observed for the preceding
amino acid (C'$_{i-2}$). However additional peaks are detected also with neighbouring ones and support the
sequence-specific assignment process.
It is interesting to note, once the sequence-specific assignment becomes available, that C'$_{i-1}$-N$_i$ correlations
fall in distinctive spectral regions of the CON$^{Pro}$ 2D spectrum, as already pointed out for selected residue
pairs such as Gly-Pro, Ser-Pro, Thr-Pro, Val-Pro (Murrali et al., 2018). For example, inspection of Figure 1
allows us to identify Gly-Pro pairs in all the CON spectra of different proteins from their characteristic
chemical shifts (in the top-right portion of the proline region). An additional contribution towards smaller
$^{15}N$ chemical shifts can also be identified in cases in which more than one proline follows a specific amino
acid-type such as for Ala-Pro, Met-Pro and Gln-Pro cross peaks that are shifted to lower $^{15}N$ chemical shifts
when an additional proline follows in the primary sequence. These effects are likely to originate from a
combination of effects deriving from the covalent structure (primary sequence in this case) as well as from
local conformations. Needless to say that the experimental investigation of these aspects in more detail
constitutes an important point to describe the structural and dynamic properties at atomic resolution of





the proline-rich parts of highly flexible IDRs. The proposed experiments are thus expected to become of
general applicability for studies of IDPs/IDRs in solution.
The data generated on proline-rich sequences are of course very relevant to populate the BMRB with
more information on proline residues in highly flexible protein regions. This in turn will generate more
accurate reference data in chemical shift databases to determine local structural propensities through the
comparison of experimental shifts with reference ones (Camilloni et al., 2012; Tamiola et al., 2010)
improving our understanding of the importance of transient secondary structure elements in determining
protein function. On this respect, CBP itself provides another enlightening example with CBP-ID3 (residues
674-1079 of CBP), which features a high number of proline residues (75 out of 406 residues) representing
18% of its primary sequence. The distribution in this case is along the entire sequence but less frequent
toward the end, where a β-strand conformation propensity is sampled.  Also in this case the distribution
of proline residues is important in shaping the conformational space accessible to the polypeptide,
facilitating the interaction with protein's partners (Contreras-Martos et al., 2017).
Determination of additional observables, such as the $^3J_{C'C'}$ through the 3D (HCA)COCON$^{Pro}$ experiment or
of different ones through modified experimental variants of these experiments are expected to contribute
to the characterization of novel motives in IDRs/IDPs.
The experimental strategy proposed here focuses on a remarkably small spectral region which however
turns out to be one of the most interesting ones, in particular in the perspective of studying IDPs/IDRs of
increasing complexity, somehow reminiscent of other strategies that have been proposed in which only
selected residue types are investigated to access information on challenging systems (such as for example
the studies of large systems enabled by Methyl-TROSY spectroscopy (Kay, 2011; Schütz and Sprangers,
2020). Interestingly the analysis of the NMR spectra presented here enables one to classify the observed
cross peaks into residue types, also in absence of sequence-specific assignment. This might provide
interesting information for complex IDPs/IDRs in which one is interested in the investigation of the
contribution of specific residue types, such as to monitor the occurrence of post-translational
modifications or even other phenomena that are more difficult to investigate like liquid-liquid phase
separation.



**Conclusions and perspectives**

Detection and assignment of proline-rich regions of highly flexible intrinsically disordered proteins allows us to have a glimpse on the ways in which proline residues encode specific properties in IDRs/IDPs by simply tuning their distribution along the primary sequence. NMR spectroscopy is particularly well suited for the task, since proline residues have attractive features from the NMR point of view, starting from the peculiar chemical shifts of $^{15}$N nuclear spins. In addition, the lack of the attached amide proton implies that one of the major contributions to relaxation of $^{15}$N spins is absent and thus proline nitrogen signals have small linewidths. These characteristics make them a very useful starting point for sequential assignment purposes and structure characterization. Furthermore, they provide a set of NMR signals with promising properties to enable high-resolution studies of increasingly large IDPs/IDRs.

Several approaches either based on H$^N$ or on H$^\alpha$ direct detection have been proposed to bypass the problem introduced in sequence-specific assignment by the lack of amide protons typical of proline residues (Hellman et al., 2014; Kanelis et al., 2000; Karjalainen et al., 2020; Löhr et al., 2000; Mäntylahti et al., 2010; Tossavainen et al., 2020; Wong et al., 2018). While these can result useful for systems with moderate complexity (H$^\alpha$ detection) or for systems that feature isolated proline residues in the primary sequence (H$^N$/H$^\alpha$), they are not as efficient for complex IDRs/IDPs in which high resolution is mandatory and in which often consecutive proline residues are encountered. Thus, the proposed experiments are crucial to assign intrinsically disordered protein regions presenting many repeated motives including proline residues as well as poly-proline segments.

Since NMR probeheads with high $^{13}$C sensitivity have become widely available, it is expected that this set of experiments will be applied as an easy-to-use tool also to complement H$^N$-based assignment.

**Acknowledgments**

The support of the CERM/CIRMMP center of Instruct-ERIC is gratefully acknowledged. This work has been funded in part by a grant of the Fondazione CR Firenze and by the Italian Ministry of University and Research (FOE funding). Maria Grazia Murrali, Letizia Pontoriero and Marco Schiavina are acknowledged for their contribution in the early stages of the project.



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
