# Peer review of "Exclusively heteronuclear NMR experiments for the investigation of intrinsically disordered"

_Magnetic Resonance, 2021_

## Author Comment (AC1)

**RC1:**

This is an interesting manuscript showing application of Proline selective NMR experiments in intrinsically disordered proteins. The topic is important since prolines are abundant is this kind of proteins. To achieve the goal the direct 13C detection is used, where the lower sensitivity is compensated by lack of solvent exchange, better dispersion, and direct access to proline residues. The experiments are convincingly demonstrated. The pulse-sequences are described carefully, thus it should allow for repetition of reported results. Authors provide several examples of successful applications and consider their utility.

The article is well and clearly written, and is certainly interesting for Journal readership. Therefore, I am recommending the paper for publication after a minor revision, see below for details:

The informative value of the article certainly could be improved after adding of a few sentences regarding band-selective homodecoupling instead of IPAP and possible 15N detection.

We thank the reviewer for the positive evaluation of our manuscript.

As suggested by the referee, we have included a comment in the "Conclusions and perspectives" section on the use of $^{15}$N-detected experiments for proline residues assignment, with particular reference to the Gerhard Wagner's PNAS paper published in 2018. The same article has been cited also in the "Introduction" as an example of articles reporting proline residues assignment.

Alternative approaches to achieve homonuclear decoupling in the direct acquisition dimension have been mentioned when describing the pulse sequence implementation (Figure 3). These methods are certainly useful when an additional element should be appended to the pulse sequence for the implementation of the IPAP approach, introducing additional delays and pulses. However many experiments end with a coherence transfer step in which antiphase carbonyl operators are refocused to in-phase ones and in these cases the IPAP approach can be implemented by simply shifting the position of pairs of Ca 180° pulses the final coherence transfer step. This is the case of the CON experiment as well as of the 3D experiments based on carbonyl direct detection for sequence specific assignment. Therefore the implementation of other approaches for homonuclear decoupling is not such a stringent requirement in these cases.

**RC2:**

In this work the authors present a simple to implement modification of the $^{13}$C,$^{15}$N CON detection platform that is optimized to selectively record correlations from proline nitrogens with high resolution and short acquisition times. There is interest in this development because highly proline-enriched intrinsically disordered proteins (IDPs) remain challenging to study by NMR. Carbon direct-detect spectroscopy has arisen to meet this area of need and this work is a welcome step toward greater efficiency. Although this project reports a minor step forward from the author's prior work (Murrali 2018 in the references) novel pulse programs are presented that were only conjectured in the discussion of the previous work and the contextualization of this method is more fully developed. Thus, this work is well aligned with the editorial scope of the journal and should be of use to the field. Minor suggestions for revision follow.

The authors and their past/present collaborators have always been careful to reserve the term "protonless" in reference to NMR experiments for those in which the coherence transfer pathway neither begins nor ends with the $^1$H nucleus. Here, the authors present the strategy as being "protonless" and yet many of the pulse sequences utilize the proton-start strategy for higher sensitivity. Perhaps this is semantic, but a bit of clarification in the introduction may be helpful.

Table 1 is a very helpful guide for readers, but it did raise a question in my mind. Why are the carbon spectral widths set to 30 ppm for carbonyl (F2 in 2D and F3 in 3D)? This seems overly broad for carbonyl.

A critic might point out that these experiments are redundant for most systems; the regular CON already contains the proline resonances and coupling the CON-Pro to HSQC triple-resonance assignments offers completeness, but it probably could have been achieved with purely carbon-detected strategies anyway. However, this criticism misses the major point so carefully outlined by the authors in the discussion and conclusions: these experiments are excellent for exceptionally large or otherwise complex to deal with IDPs, filling a role analogous to methyl selective labeling and other strategies used for large folded proteins. My question and comment therefore come down to the text beginning on line 236, which states "the initial count of cross-peaks in this spectrum reveals 42 out of the 45 expected correlations, highlighting the potential of this experimental strategy for the investigation of IDRs/IDPs of increasing complexity" and the associated spectrum in Figure 4B. How much better is the resolution in the 2D CON$^{Pro}$ compared to what would be achieved in the traditional 2D CON? How many peaks are resolvable with the new technology that are overlapped or ambiguous with the original CON? If it wouldn't be too distracting it seems that an overlay or some other visualization of the improvement on offer for a complex IDP like CBP-ID4 may be helpful for readers.

This is very minor, but on line 329 it seems "motives" should be replaced with "motifs".

We thank the reviewer for the positive evaluation of our manuscript.

As pointed out correctly by the reviewer, the term "protonless" refers to experiments in which protons are not perturbed in any of the magnetization transfer pathway. In this paper we prefer to opt for the experimental variants that exploit $^1$H polarization as a starting source to increase experimental sensitivity. These experiments are generally referred to as "exclusively heteronuclear NMR experiments" to indicate that chemical shifts are exploited in the dimensions of NMR experiments, including the acquisition ones, but that they are not "protonless" because $^1$H is used as a starting polarization source (as now indicated more explicitly in the introduction in page 2, line 53).

Concerning the spectral width on C', it is true that 30 ppm is large! However, we just kept the acquisition time constant and shorter than 100 ms to avoid damage to the probehead due to simultaneous decoupling of $^1$H and $^{15}$N nuclei. Of course a smaller spectral width could be used reducing in parallel the number of points to keep the acquisition time within the 100 ms time limits for safe probe operation. By using more gentle decoupling or a more robust probehead, the acquisition parameters can be changed accordingly.

Concerning the final comment on resolution, similar results can be obtained using the regular CON and the CON(pro) experiments, provided the number of points acquired in the indirect dimension is properly set. The point of using these proline-tailored experiments is two-fold: 1) simplify the spectral complexity and 2) save experimental time in case sensitivity is not a limiting factor. A comment has been added in the "Conclusions and perspectives" section (page 15, line 338). Thank you for stimulating this clarification.

The typo has been corrected.

---

## Author Response (AR2)

The codes of the pulse sequences used have been uploaded as a Supplement.

A statement has been introduced in the final part of the manuscript:

**Supplement**

The codes of the pulse sequences used are reported as Supplementary Information. Data are available upon request to the authors.